# Does Vaccinating against Influenza in a Given Epidemic Season Have an Impact on Vaccination in the Next Season: A Follow-Up Study

**DOI:** 10.3390/ijerph19137976

**Published:** 2022-06-29

**Authors:** Maria Ganczak, Paulina Dubiel, Marzena Drozd-Dąbrowska, Marcin Korzeń

**Affiliations:** 1Department of Infectious Diseases, University of Zielona Góra, Zyty 28, 65-046 Zielona Góra, Poland; pdubiel@uz.zgora.pl; 2Primary Care Clinic, Parkowa 7, 74-100 Gryfino, Poland; marzena.dd@interia.pl; 3Department of Artificial Intelligence and Applied Mathematics, West Pomeranian Institute of Technology, Zolnierska 48, 71-210 Szczecin, Poland; mkorzen@wi.zut.edu.pl

**Keywords:** vaccination, influenza, intentions, coverage, predictors, elderly, follow-up study

## Abstract

To improve the uptake of influenza vaccine in the elderly, it is important to understand the factors that predict vaccination. The study objective was to explain influenza vaccination uptake in the next season (2019/2020) in a sample of primary care clinic patients from Gryfino, Poland, vaccinated in 2018/2019 with the free-of-charge quadrivalent vaccine. A baseline and a follow-up survey assessed respondent intentions to receive a vaccine (2018), then (2020) vaccine uptake and its predictors. Patients (*n* = 108, 54.6% males, M_age_ = 66.7 ± 6.7) filled in a researcher-administered questionnaire. A majority (69.3%) intended to get vaccinated in the next season, with 25.9% receipt. Of those willing to be immunized, only 31.9% were vaccinated in the next season; of those whose decision was dependent on reimbursement, none received influenza vaccine; of undecided patients, 23.1% were vaccinated. Multivariable analysis indicated that living with a partner (OR 6.22, *p* = 0.01), being employed (OR = 4.55, *p* = 0.05) and past vaccination behavior (OR 4.12; *p* = 0.04) were predictors of vaccine uptake. The findings show limited follow-through on initial influenza vaccination plans for the nearest season in previously vaccinated elderly patients. Future interventions should additionally focus on unanticipated barriers to vaccination, such as those revealed in this study, to increase vaccination coverage rates.

## 1. Introduction

Results of the Burden of Communicable Diseases in Europe study showed influenza having the highest burden among all infectious diseases in Europe between 2009 and 2013, with 81.8 disability-adjusted life years (DALYs) per 100,000 persons, 29.8% of total DALYs [1]. The elderly are among the acknowledged high-risk groups because they suffer from increased morbidity and mortality from seasonal influenza infection [1,2,3].

Although several infection control measures can help prevent seasonal influenza, vaccination is considered the most effective [4]. In Poland, according to the National Immunization Program, vaccination with the quadrivalent influenza vaccine (QIV) (with seasonal antigen combination) is currently recommended for children 0.5–18 years, patients > 55 years, with high-risk conditions, as well as for those with increased risk for transmission [5]. Recommendations of the Polish Advisory Committee on Immunization regarding the use of seasonal influenza vaccine are updated on an annual basis. There are no special referrals to get the influenza vaccination; however, vaccination sites are operated on an appointment-only basis. Between the 2018/2019 and 2019/2020 seasons, the cost of a vaccine was entirely covered by a patient, except for patients > 65 years, for whom it was 50% refunded. However, totally refunded vaccination for senior citizens was exclusively provided within some programs financed by local authorities. Between September 2020 and November 2021, the free of charge QIV was offered regarding patients aged 75 years and over, pregnant women, those in long-stay residential care homes, their caregivers, frontline medical staff, medical students, and those exposed to contact with a large number of individuals, including persons working in education, trade, army, and transport sectors. Since 23 November 2021, the free-of-charge QIV has been offered for all adults via national health insurance; money has not been advanced [6]. From winter 2022 to 2023, Polish citizens will not be offered a free of charge influenza vaccine through the National Health Fund.

Although the WHO urged to implement adequate strategies to increase influenza vaccination uptake of high-risk persons, with the goal of reaching 75% vaccination coverage, in many European Union (EU) countries, including Poland, coverage for aging populations still remains below the WHO recommendations. According to the latest report, in 2020, influenza vaccination uptake in Poland in those aged ≥65 years was 7.67% [7] and was one of the lowest in the EU [8].

In the countries with disturbingly low vaccination coverage for the elderly, such as Poland, for immunization to be a workable strategy, nonetheless, senior patients’ vaccination beliefs and actions should be better understood. Previous studies have explored patient-level determinants associated with acceptance of influenza vaccination uptake among the elderly, including demographic factors, such as being older, married, non-smoker, of a higher social class, higher education, a higher household income, health insurance, as well as having poor self-assessed health, a chronic illness, and having a family doctor [9,10,11,12]. Lack of knowledge about influenza, inadequate physician–patient relationships, and concerns about vaccine effectiveness can also influence vaccine hesitancy and were also found to have a detrimental impact on poor vaccine coverage [10,11].

The recommendation of influenza vaccination by the physician can motivate vaccine uptake [11,13,14]. Most vaccinated patients report that their decision was based on a family doctor’s recommendation [11,13,15]. Furthermore, receiving relevant information is a strong determinant for planning on being vaccinated the next year [9,11,13].

Effective communication about influenza-related issues, including vaccines and vaccination, requires clear-cut strategies for recognizing which patients are most and least predisposed to be vaccinated [16]. Such information can assist medical staff in focusing specifically on unvaccinated patients who are most likely to refuse an influenza vaccine. However, previous evidence from non-pandemic influenza seasons shows sizable differences between patient-stated vaccination intentions prior to vaccine availability and later vaccine uptake, indicating a significant lack of follow-through on initial immunization plans [16,17,18,19,20,21]. Most of the above-mentioned follow-up studies assessed how vaccination intent translates into immunization during the current season, surveys which evaluated how to predict and explain influenza vaccination uptake in the next season are rather scant.

Therefore, the study objective was to determine whether vaccinating patients against influenza with a refunded vaccine in a given epidemic season (2018/2019), together with education about influenza, has an impact on vaccination in the following season (2019/2020). Factors influencing the continuation of vaccination in the next season were also determined. Such data could help to better develop adequate strategies to increase vaccination coverage among the elderly.

## 2. Materials and Methods

### 2.1. Study Population and Setting

A baseline survey assessed respondent demographics and intentions to receive a vaccine. This was conducted among consecutive patients attending a vaccination point at one primary care clinic (PCC) located in the city of Gryfino, Poland, between October and December 2018 to get a refunded quadrivalent influenza vaccine (QIV); the cost of the vaccine was covered by the Mayor of the City. The city is located in the north-western part of Poland; the recent census recorded a 20,792 population in 2021 [22]. Patient inclusion criteria were as follows: age ≥ 55 years, lack of co-existing diseases that could affect cognitive functions, lack of contraindications to immunization, and informed written consent. Medical examination was undertaken before the patient’s immunization; this was combined with short education about the benefits of the QIV; every subject received an information leaflet on influenza downloaded from the website of the National Influenza Control Program (www.opzg.pl accessed on 27 May 2022) [23].

### 2.2. Study Instrument

A questionnaire (Appendix A) was designed by the study team using a literature review [9,10,11,12,16,17,18,19,20,21] and was then administered to participants by the medical staff. Before fulfilling this questionnaire, patients were informed about the aim of the study and then assured that they would not be identified in any presentation or publication. In case of any doubts in understanding the questions, respondents could ask a research team member present at the place about how to complete the survey. The time of completing the questionnaire was about 10 min.

The questionnaire contained 26 questions and was composed of 3 sections: (1) demographic data and medical history (age, gender, marital status, employment, comorbidities); (2) data on influenza vaccination (previous sources of information, reasons for vaccination, previous immunizations, adverse effects, intent to vaccination in the next season); (3) knowledge about influenza (1 question about sources of information, and 13 questions assessing the knowledge level). To evaluate the understanding of questions, a pilot study was conducted among patients > 60 years old admitted to the local hospital. Twenty patients were interviewed; collected data were not included in the study.

Knowledge of influenza was assessed by giving 1 point for each correct answer to the 13 items rated as “true”/“false”/“do not know”, divided into the following groups: epidemiology; clinical manifestations and complications; transmission; preventive measures, including vaccination. The scale measured knowledge from a minimum of 0 to a maximum of 13. Scores for individuals were summed up to give a total knowledge score. Scores of 0–7 (≤55% of correct answers) were arbitrary taken as poor, 8–13 (more than 55% of correct answers)—as adequate knowledge.

### 2.3. Follow-Up Study

A follow-up survey at the end of the next season assessed vaccine uptake and its predictors. All patients who participated in the 2018 study were followed for the next 15–18 months. In March 2020, they were contacted by the research team member and asked about influenza immunization in the recent (2019/2020) season. The final result was incorporated into the study database and analyzed.

### 2.4. Statistical Analysis

Data analysis was carried out with STATISTICA (PL Version 12.1, StatSoft Inc., Kraków, Poland, 2005) and R software [24]. Categorical data were presented as frequencies with percentages and continuous data as means. The endpoint variable was immunization for influenza in the 2019/2020 season after being immunized in the 2018/2019 season; we aimed to identify determinants associated with this endpoint. The bivariate analysis included variables such as age (≤67 years/>67 years), gender, source of income (employment/pension, disability benefit dependent, no income), marital status (alone, including divorced, widowed, in separation, living with a partner), together with comorbidities (yes/no), recommendation of influenza vaccination (GP-nurse/other), influenza vaccination in the previous season (2017/2018), influenza vaccination ever before the 2017/2018 season, willingness to be vaccinated in the next season (yes/no/do not know), knowledge level about influenza (adequate/poor), associated with our outcome variable. For categorical (binary) variables, as described above, groups were compared using the chi square and Fisher tests. To build a logistic regression model [25], the set of predictors was used with the help of the R MASS package [26]. Final associations between predictors and the outcome adjusted for covariates were measured with the use of coefficients of a logistic regression model. Coefficients for binary variables are equal to the natural logarithm of the odds ratio.

## 3. Results

### 3.1. Demographic Characteristics of Patients

Among 121 invited patients, 108 accepted to participate in the 2018 study, giving a response rate of 89.3%. There were no statistically significant differences between 108 patients who participated and 13 patients who did not regarding basic socio-demographic characteristics, such as gender, age, comorbidities, and influenza vaccination in the previous season. More than half of the respondents were males (54.6%). The age of participants ranged from 55 to 85 years (mean: 66.7 ± 6.7 years). About two-thirds of participants were living with a partner (married: 65.7%, cohabitant: 1.9%), 13% were widows/widowers, 10.2% were divorced 8.3% defined themselves as “maiden/bachelor”. In total, 68.5% of respondents were retired, 14.8% were still working, 10.2% ticked “disability sickness allowance”, and 6.5% did not define their employment status. More than one-third (34.3%) reported comorbidities, mainly diabetes (25.0%), followed by cancer (7.4%), autoimmune disease (4.6%), and renal failure (3.7%).

### 3.2. Influenza Knowledge Scores

The mean score of knowledge about influenza was 6 ± 2.74. Only 28.7% (31/108) of respondents scored > 55% of the correct answers. Regarding knowledge that influenza is a viral disease, less than half (46.3%) gave the correct answer. Most participants (83.3%) did not know the annual mortality burden of influenza, and 61.1% did not know that influenza A viruses infect a wide variety of animals. More than three-thirds (67.6%) correctly stated that influenza and the common cold are caused by different viruses and that influenza can spread even before infected people show symptoms; 81.5% knew that influenza is a potentially life-threatening illness; only one-third (33.3%) correctly identified people 65 years and older as being at higher risk of developing complications from influenza. When asked about vaccination, only 25.9% knew that some people have contraindications to influenza vaccines; however, 85.2% correctly recognized that vaccination should be administered annually to provide optimal protection against infection, and 81.5% stated that the best time to receive a vaccine is by the end of October.

### 3.3. Influenza Vaccination Status

Prior to the 2017/2018 season, one-third of participants (41/108; 33.3%) took the influenza vaccine at least once in the past. Only 15.7% of respondents reported being vaccinated one season before the study was conducted (2017/2018); Figure 1.

Patients vaccinated during the 2018/2019 epidemic season were asked about the reasons for being vaccinated. The results are presented in Figure 2. The most commonly reported reason was the recommendation of a family physician (46.3%), followed by the vaccination reimbursement (28.7%), recommendation of a vaccinated friend/family member (12.9%), or media campaign (12.0%). It was a multiple-choice question.

The willingness to be vaccinated against influenza in the next epidemic season (2019/2020), regardless of whether the vaccination will be refunded, was declared by 63.9% of respondents; 9.3% would make the decision to vaccinate dependent on the fact of reimbursement, only 3 out of 108 respondents answered “I am not planning to be vaccinated in the next season”, and 24.1% were undecided; Figure 3.

All patients who had been vaccinated against influenza in the 2018/2019 season were then followed until the end of the next epidemic season (2019/2020). Overall, around one-quarter (25.9%) reported they received immunization in the 2019/2020 season. Figure 4 shows the declared intent of immunization in the next season by receiving influenza vaccination in this next season. Of 69 patients with a willingness for immunization, only 22 received the influenza vaccine in the next season; of 10 patients who made their decision dependent on the reimbursement, none received the influenza vaccine in the following season. Of 26 undecided patients, 20 were not vaccinated in the next season. Almost two-thirds of the participants (63.0%) had never been vaccinated against influenza, 13.9% were vaccinated only once in their lifetime, and 23.2% more than once.

### 3.4. Factors Associated with Continued Adherence to Influenza Vaccination

Factors associated with continued adherence to influenza vaccination in the next epidemic season (2019/2020) among patients vaccinated in the 2018/2019 season with a non-paid vaccine were assessed. The results are presented in Table 1. The fraction of vaccinated patients was significantly higher among those living with a partner (*p* = 0.002), still employed (*p* = 0.03), vaccinated in the 2017/2018 season (*p* = 0.002), and ever vaccinated (*p* = 0.006). There were no statistically significant differences between vaccinated and unvaccinated patients in terms of age, gender, prevalence of comorbidities, person who recommended the QIV vaccination, willingness to vaccination in the next season, or level of knowledge about influenza.

### 3.5. Predictors of Being Vaccinated in the Following Year

Multiple logistic regression analysis regarding an association of being vaccinated in the following season with selected variables revealed that living with a partner and being vaccinated in the previous influenza season were independent positive predictors of completing the influenza vaccination (OR 6.12; *p* = 0.01 and OR 4.22; *p* = 0.04, respectively); Table 2. Being unemployed was associated with lower odds for vaccination receipt (OR 0.22; *p* = 0.05).

## 4. Discussion

### 4.1. Results Overview

To our knowledge, this was the first follow-up study in Poland and in Central Europe that surveyed the elderly PCC patients, recipients of the free of charge QIV during the 2017/2018 influenza season, to assess their intentions related to influenza vaccination, as well as vaccine receipt in the next season.

Although about three-thirds of ≥55-year-old patients immunized against influenza in a given season (2018/2019) were willing to be vaccinated in the next season, only a quarter of them received immunization (2019/2020). The study showed that among the demographics we collected from participants, living with a partner and being employed were predictive of receipt in the next season. Additionally, a report of past receipt of the influenza vaccine was also predictive.

### 4.2. Reported Influenza Vaccination Intentions and Vaccine Receipt in the Next Season

In line with some previous studies conducted in other countries, such as the US, Canada, the Netherlands, Spain, Tunisia, and China [16,17,18,19,20,27], this study found a sizable gap between reported influenza immunization intentions and vaccine receipt in the next season. For instance, 64.7% of elderly patients with chronic diseases in Tunisia expressed a willingness to be vaccinated in the next season [27]. Despite the fact that 64% of our patients had the intention to get influenza vaccination in the following season, the vaccination uptake during the next season was very low (26%). However, it was about 10% higher compared to the uptake reported by this group in the previous season and more than three times higher than reported in 2020 in Poland in those aged ≥65 years [7].

In some other EU countries with a high seasonal influenza vaccination coverage in the general population, people vaccinated against influenza in one season tend to be vaccinated in the following one [8,17]. For instance, in a Spanish study of patients vaccinated against influenza in the 2009–10 season, 87% were vaccinated in the 2010–11 season [17]; this is more than three times higher uptake than reported in our study.

The results of our survey show that addressing influenza vaccine hesitancy remains a challenge. The reasons behind it are rather complex [28,29]. All our patients who initially presented as unwilling to be vaccinated remained unvaccinated in the next season; vaccine receipt among undecided participants was only 23%. Nevertheless, vaccine-hesitant patients who were on the fence far outnumbered vaccine refusers; therefore, counseling this group might be more effective [19].

Reported past receipt of influenza vaccine appears to be one of the strongest predictors of influenza vaccination receipt in older patients. Similar results were obtained by other authors. For instance, Ye et al., who conducted a follow-up survey among diabetic patients in China found that vaccination history displayed positive associations with influenza vaccine uptake [18]. The results of a follow-up survey at the conclusion of the influenza season, which assessed self-reported influenza vaccine uptake among Canadian pregnant women, showed that last year’s vaccination behavior predicted vaccine uptake [19]. The results of all above-mentioned studies show that if patients have positive initial influenza immunization experiences, they are likely to receive vaccination regularly and get “into the habit” of being immunized [13,18,19,27].

Our study indicated that living with a partner was a predictor influencing continued adherence to influenza vaccination in elderly persons vaccinated in the preceding season. Although marital status has not been assessed by other authors regarding factors affecting the continuity of vaccination, several publications systematically reviewed barriers associated with influenza vaccination in a given season among older people and found that living alone was related to a significantly decreased vaccine uptake [9,10,11,12]. For instance, compared with not living alone, living alone was associated with a 30% decrease in influenza vaccination uptake in Europe [12]. The authors concluded that, in general, living alone is associated with the worst care experience and immunization among older people [30].

The results of our study showed that, in elderly persons, not being employed was a barrier to influenza vaccination in the next season. Of note, 68.5% of the respondents were retired; according to the current survey, around two-thirds of Poles assume that pension is not enough to live even at a minimum level [31]. This may be one of the important reasons contributing to the low influenza uptake in the 2019/2020 season, during which our respondents did not have a chance to benefit from free-of-charge vaccinations. Similarly to our findings, other authors also reported a higher uptake of influenza vaccine for individuals with a higher income [9,11,12].

In the study conducted by Ye et al., the payment of influenza immunization was one of the most essential factors affecting later vaccine uptake [18]. In Poland, in the 2019/2020 season, when the study was conducted, the cost of the vaccine was about 30 PLN, which was acceptable for the working class; however, it could have been a problem for the elderly, retired people with the low income. Notably, none of the studied patients vaccinated by the free of charge QIV in the 2018/2019 season—who declared that their decision to vaccinate in the next season would be dependent on reimbursement—received the vaccine in the 2019/2020 season.

The price of a vaccine may deter the elderly, specifically retirees, from acquiring the vaccine. Thus, there is a need for the QIV vaccine to be provided without any cost for this age group. The provision of free-of-charge influenza vaccination on a national level could be a driver for increasing vaccination uptake. This is particularly important in the context of an aging Polish population. Eurostat has recently reported that Poland is in the second position among EU countries with the highest increase in the share of the population aged ≥65 years between 2011 and 2021 [32]. Influenza causes a heavy disease burden, especially in high-risk populations, such as older adults, but can be effectively alleviated by immunization [33,34]. Brydak et al., assessed the cost-effectiveness of the full reimbursement of an influenza vaccination program in Poland for people aged ≥65 years and found that the implementation of such a program by the National Health Fund would be a cost-effective strategy [35]. Results of some other current studies also support the implementation of a government fully funded older adult vaccination program [33,34,36,37].

In fact, since November 2021, free influenza vaccines have been provided in Poland to adults, including people aged 60 years and above. Further studies are needed to assess to which extent this vaccination policy could increase the vaccine uptake rate among the elderly.

Although—in contrast with some other publications [18]—we did not observe the association of a family physician recommendation with actual influenza vaccine receipt, medical staff might have a positive effect on our patients’ vaccination. Rather than working directly on the actual vaccine uptake, the recommendations from medical professionals might improve patients’ awareness of influenza susceptibility. As mentioned previously, in this study, influenza vaccination in a given season was combined with short education about the benefits of the QIV provided by a family physician who also distributed information leaflets explaining why the vaccine is being offered and how, when, and where it is given.

Of note, most vaccinated elderly patients surveyed by us previously [13] reported that their decision was based on a family doctor’s recommendation. As illustrated by our previous study results, receiving information on influenza vaccination was correlated with five times higher likelihood of being immunized. Moreover, receiving relevant information was a strong determinant for planning on being vaccinated the next year [13,18,19]. Regarding the current study participants, the most commonly (46.3%) reported reason for influenza vaccination in the 2018/2019 season was the recommendation of a family physician. As a trusted source of information, medical staff may play a key role in driving vaccine uptake.

In contrast with some other studies’ results, we did not find age, gender, chronic conditions, or knowledge to be related to continued adherence to influenza vaccination [17].

### 4.3. Limitations

A key limitation of the current study was the small sample size. This was due to the high cost of serological tests required to assess immunity, performed in all 108 patients before and after immunization (2018/2019 season) [23]. Second, this was a convenient sample of consecutive patients recruited by one PCC; therefore, this may not adequately represent the real population of the elderly patients due to selection bias and may limit the generalizability of findings. It was possible that patients who were more concerned about health were more likely to participate in the survey. Third, other unmeasured factors, such as attitudes and social norms and the beliefs on which these are based, could have also influenced adult vaccination intentions and behavior [19,38]. Finally, due to the lack of the Polish national/regional immunization register, it was impossible to check the patient’s influenza vaccination status in previous seasons through medical records. For the purpose of this study, it was obtained either through a vaccination record presented to the research team or through an adequate answer in the study questionnaire. Recollecting information about past receipt of influenza vaccine could introduce a recall bias. Further studies on larger populations are needed to better assess differences between patient-stated vaccination intentions prior to vaccine availability and later vaccine uptake.

### 4.4. Implications for Practice

Concerning clinical practice, our results indicate that regarding the elderly PCC patients who have been vaccinated against influenza in a given season with a free of charge vaccine, only around one-fourth continue immunization in the next season; less than half of those who declare to vaccinate in the next season receive a vaccine. This is in line with the results reported by Harris et al., who surveyed American adults, specifically recommended by the ACIP for vaccination against seasonal influenza, and found that over half who intended to be vaccinated had been vaccinated [16]. Our findings show that screening this group of patients on the basis of demographic characteristics, such as marital and employment status, could be effective in predicting influenza vaccine uptake. Additionally, checking whether a patient regularly gets influenza immunization would better help a physician differentiate between those expected to receive the vaccination without further encouragement (i.e., vaccinated on a regular basis) and those potentially requiring extra time and more information on the risks and benefits of the product [19]. If a patient lives alone, is retired, and is known not to have received influenza immunization in the recent past, further discussion would be required to address potential vaccine hesitancy in order to positively change attitudes regarding vaccination.

## 5. Conclusions

In conclusion, the findings show limited follow-through on initial influenza vaccination plans for the nearest season in previously vaccinated elderly patients. This points toward unanticipated barriers to vaccination. Future interventions should focus on the strongest predictors of influenza vaccination receipt identified in this study to increase vaccination coverage rates.

## Figures and Tables

**Figure 1 ijerph-19-07976-f001:**
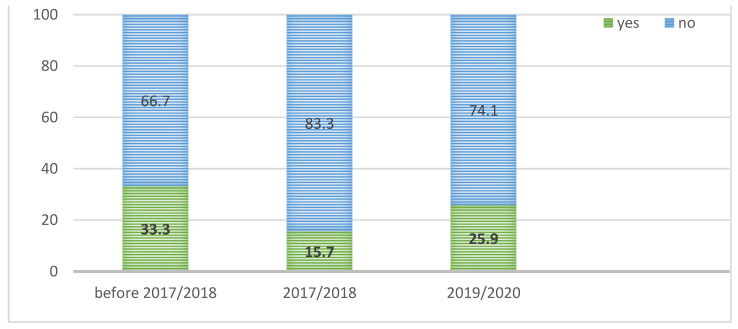
Vaccination history among the surveyed patients by the epidemic season; (*n* = 108).

**Figure 2 ijerph-19-07976-f002:**
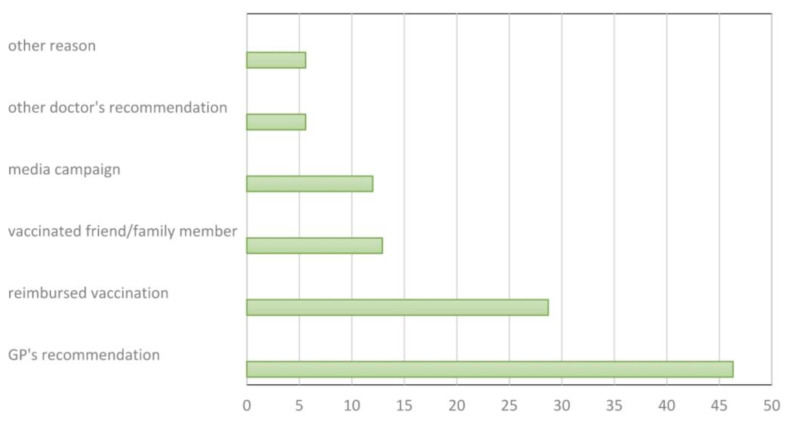
Reasons for influenza vaccination in the 2018/2019 season; Gryfino, Poland (*n* = 108).

**Figure 3 ijerph-19-07976-f003:**
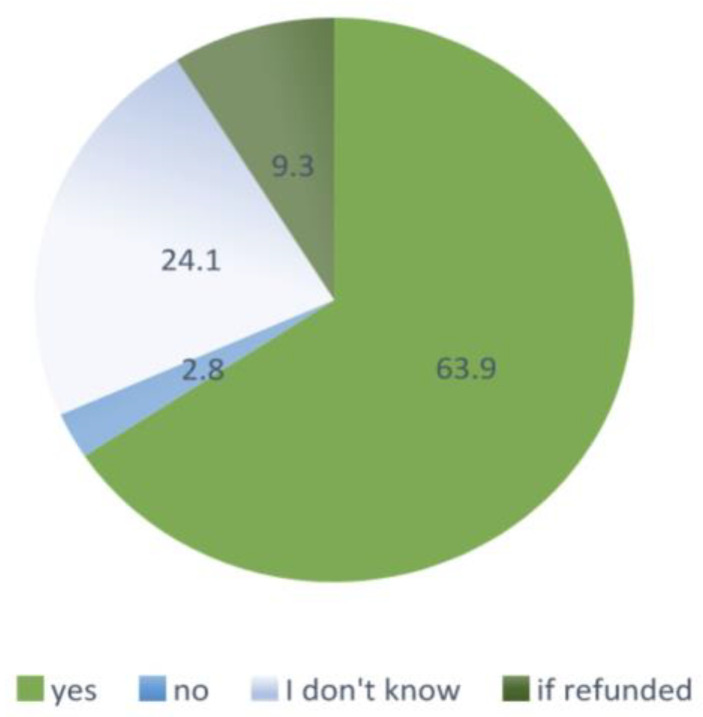
Intentions to vaccinate against influenza in the next epidemic season (2019/2020); Gryfino, Poland (*n* = 108).

**Figure 4 ijerph-19-07976-f004:**
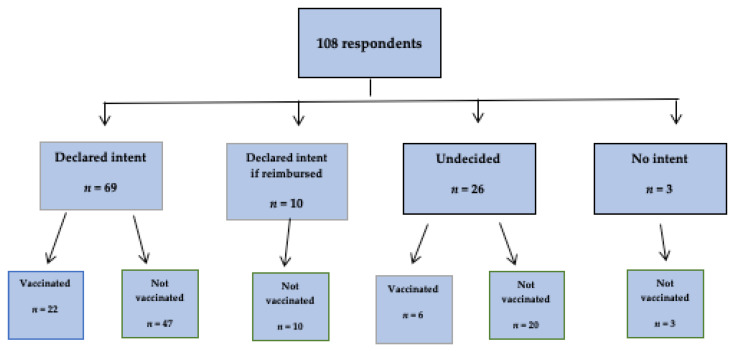
Influenza vaccination intent in the next season (2019/2020) by immunization receipt; Gryfino, Poland (*n* = 108).

**Table 1 ijerph-19-07976-t001:** Factors affecting the continuation of vaccination in the epidemic season.2019/2020, Gryfino, Poland; (*n* = 108).

Characteristics		*n*/N	%	*p*	OR	95%CI
Gender	Female Male	13/49 15/59	26.5 25.4	1.00	0.59	0.41–2.74
Age (years)	55–67 68–85	18/59 10/49	30.5 20.4	0.27	1.06	0.21–1.54
Marital status	Married Other	25/71 3/37	35.2 8.1	0.002 *	6.07	1.64–33.95
Employed	Yes No	8/16 20/92	50.0 21.7	0.03 *	3.57	1.02–12.5
Comorbidities	Yes No	9/37 19/71	24.3 26.8	0.82	0.88	0.31–2.38
Vaccinated in the 2017/2018 season	Yes No	10/17 18/91	58.8 19.8	0.002 *	5.67	1.69–20.28
Vaccinated ever before the 2017/2018 season	Yes No	17/41 11/67	41.5 16.4	0.006 *	2.61	0.98–7.02
Who recommended vaccination	Doctor/nurse Other	16/57 12/51	28.1 23.5	0.66	1.19	0.23–4.16
Vaccination intent in the next year	Yes No	22/69 6/39	31.9 15.4	0.07	1.47	0.49–5.03
Sources of knowledge about influenza	Doctor/nurse Other	17/71 11/37	23.9 29.7	0.64	0.75	0.28–2.03
Influenza knowledge level	High Low	9/31 19/77	29.0 24.7	0.82	1.25	0.43–3.44

* statistically significant.

**Table 2 ijerph-19-07976-t002:** Logistic regression model: association of influenza vaccination in the following season with selected variables: estimates, 95% confidence intervals (CIs), and *p* values; *n* = 108.

Variable	Estimate	95% CI	*p*
Intercept	0.30	0.0005–168.61	0.71
Married/cohabitating	6.12	1.68–34.57	0.01 *
Non-employed	0.22	0.04–0.996	0.05 *
Vaccinated in the previous (2017/18) season	4.22	1.08–17.66	0.04 *
Vaccinated ever before the 2017/2018 season	1.72	0.56–5.16	0.33

* statistically significant.

## Data Availability

The data underlying this article will be shared upon reasonable request to the corresponding author.

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
