# Peer review of "Does Vaccinating against Influenza in a Given Epidemic Season Have an Impact on Vaccination in the Next Season: A Follow-Up Study"

_ijerph, 2022, doi:10.3390/ijerph19137976_

Round 1

Reviewer 1 Report

The article is interesting because it helps to understand the factors that make patients get vaccinated against influenza. It has the drawback of the small sample size, but it is well designed.

It would be interesting if the authors explained the vaccination process in more detail in the introduction so that a person from outside Poland could understand the process. It is unclear if the vaccination is free or if the patient must pay and get the money back.

The discussion seems to indicate that money has to be advanced, but this should be better explained in the introduction. The discussion contemplates that the price may deter some people (especially retirees) from acquiring the vaccine. The discussion should reflect on the need for the vaccine to be free of charge for retirees or the entire population.

Another question for the discussion and the introduction is if the bureaucratic procedures for obtaining a refund are simple or very complicated and may dissuade people from requesting a refund?

The material and methods commented that a blood draw was performed, but the information collected is not used in the article. So it could be omitted.

Another question for discussion is whether the patient's influenza vaccination status or not was obtained by interview. What is the reason? Are there records of influenza vaccinations? Why were they not consulted? Could there be any bias in using self-reported data?

Include as supplementary material the questionnaire (translated into English). It is essential for two reasons. On the one hand, it allows readers to understand the study better; on the other hand, some readers might use it or be inspired by the questionnaire when designing the study, which would lead to citations of the article.

In lines 157 to 159, the following paragraph should be deleted."

This section may be divided into subheadings. It should provide a concise and precise 158 description of the experimental results, their interpretation, and the experimental 159 conclusions that can be drawn."

The participation rate was very high. Is there any difference between those who participated and those who did not?

The results on line 165 appear (mean: 66.7±6.7 years). It should be indicated whether 6.7 is the standard deviation or standard error the first time the results are presented in this way.

Table 1 should include a column with the crude ORs and the confidence interval for each variable.

In the header of Table 2, "estimate" should be replaced by OR. The table should be self-explanatory without having to read the article.

Reviewer 2 Report

The major flaw of this study was the small size and non-representative of the study sample. There have been many studies examining the factors predicting vaccination against influenza among elder people. This study did not add new knowledge to this field.

Reviewer 3 Report

Dear Authors,

Thank you for the opportunity to review the article entitled „Does vaccinating against influenza in a given epidemic season have an impact on vaccination in the next season: a follow-up study” which  addresses factors influencing the continuation of vaccination against influenza.

The results of this study are important because they strengthens previous findings regarding the cluster of factors associated with the vaccination intent. Studies regarding this subject showed an important role of funded vaccination programmes and medical system, trust in safety and efficacy of the vaccines, health literacy, comorbidities and age. In this study, it was notice that history of influenza vaccination in the previous epidemic season and previous vaccination history were the main determinants of vaccine adherence. The questionnaire relating to influenza vaccine knowledge is very informative and along with the presence of vaccination history in a previous period represent elements of originality in this research article.

The study is correctly designed and technically sound. The methods used in this research are well described and provide sufficient details to be understand. The research methodology is in line with the proposed objectives. The results are appropriately interpreted and respond to the hypothesis of the study.

The disscutions are relevant the paper's focus area and address the findings of the research in relation with other studies and also propose potential explanation for vaccine adherence against influenza  in Poland.

The authors colected data on influenza-specific antibody titers as a correlate of protection,  which were published in a previous study. Although this is a very important aspect, it isn’t related with the aim of the present article and lines 98-100 from the Materials and Methods section should be removed.

Thank you for your esteemed efforts in increasing our collective knowledge about influenza vaccine determinants.

Sincerely yours

Alina Popa MD, PhD

Round 2

Reviewer 2 Report

The authors have added adequate explanations for this study. This manuscript can be accepted for publication.